# Efficacy of Western, Eastern, and Venezuelan Equine Encephalitis (WEVEE) Virus-Replicon Particle (VRP) Vaccine against WEEV in a Non-Human Primate Animal Model

**DOI:** 10.3390/v14071502

**Published:** 2022-07-08

**Authors:** Crystal W. Burke, Rebecca A. Erwin-Cohen, Aimee I. Goodson, Catherine Wilhelmsen, Jennifer A. Edmundson, Charles E. White, Pamela J. Glass

**Affiliations:** 1Virology Division, United States Army Medical Research Institute of Infectious Diseases (USAMRIID), Fort Detrick, Frederick, MD 21702, USA; rebecca.erwin-cohen@nih.gov (R.A.E.-C.); aimee.i.goodson.civ@mail.mil (A.I.G.); pamela.j.glass.civ@mail.mil (P.J.G.); 2Pathology Division, United States Army Medical Research Institute of Infectious Diseases (USAMRIID), Fort Detrick, Frederick, MD 21702, USA; catherine.l.wilhelmsen.civ@mail.mil (C.W.); jennifer.a.edmundson.mil@army.mil (J.A.E.); 3Veterinary Medicine Division, United States Army Medical Research Institute of Infectious Diseases (USAMRIID), Fort Detrick, Frederick, MD 21702, USA; charles.e.white174.ctr@mail.mil

**Keywords:** eastern equine encephalitis, western equine encephalitis, Venezuelan equine encephalitis, multi-agent vaccination, nonhuman primates, virus replicon particle, alphavirus

## Abstract

The purpose of this study was to evaluate the effects of the route of administration on the immunogenicity and efficacy of a combined western, eastern, and Venezuelan equine encephalitis (WEVEE) virus-like replicon particle (VRP) vaccine in cynomolgus macaques. The vaccine consisted of equal amounts of WEEV, EEEV, and VEEV VRPs. Thirty-three animals were randomly assigned to five treatment or control groups. Animals were vaccinated with two doses of WEVEE VRPs or the control 28 days apart. Blood was collected 28 days following primary vaccination and 21 days following boost vaccination for analysis of the immune response to the WEVEE VRP vaccine. NHPs were challenged by aerosol 28 or 29 days following second vaccination with WEEV CBA87. Vaccination with two doses of WEVEE VRP was immunogenic and resulted in neutralizing antibody responses specific for VEEV, EEEV and WEEV. None of the vaccinated animals met euthanasia criteria following aerosol exposure to WEEV CBA87. However, one NHP control (total of 11 controls) met euthanasia criteria after infection with WEEV CBA87. Statistically significant differences in median fever hours were noted in control NHPs compared to vaccinated NHPs, providing a quantitative measure of infection and efficacy of the vaccine against a WEEV challenge. Alterations in lymphocytes, monocytes, and neutrophils were observed. Lymphopenia was observed in control NHPs.

## 1. Introduction

Western equine encephalitis virus (WEEV) is a member of the *Alphavirus* genus of the family *Togaviridae*. The discovery and isolation of WEEV was first described in 1930. The virus can cause significant mortality in equines and humans, especially during epizootic events. During widespread epizootic events in the United States (US) and Canada during the late 1930s and early 1940s, more than 350,000 horses were infected with WEEV, accompanied by a high infection rate among humans [1]. Natural outbreaks of WEE in areas of North and South America, were the result of mosquito-borne transmission with human case-fatality rates ranging from 8% to 15% [2]. WEEV infection can also occur by aerosol, as evidenced through studies with nonhuman primates [3] and previous laboratory worker exposures, with an increased fatality rate of up to 40% [2]. This mode of transmission, as well as other properties—virus growth to very high titers, virus stability in storage, and ease of genetic manipulation in the laboratory—make WEEV a potential biological threat agent [4]. The US Centers for Disease Control and Prevention (CDC) classifies viral encephalitis viruses, including WEEV, as Category B threat agents [5].

WEEV continues to be a public health concern as both a naturally occurring virus and as a biothreat agent [6]. However, there are currently no FDA-approved vaccines for Venezuelan equine encephalitis virus (VEEV), eastern equine encephalitis virus (EEEV), or WEEV. Investigational new drug (IND) vaccines for the viruses are available for laboratory workers and others at risk of exposure [7,8]. The IND vaccine for WEEV has significant limitations that have prompted the development of second generation vaccines. A substantial benefit to public health would include new vaccine candidates that could elicit a protective response (defined as production of neutralizing antibody) against WEEV, EEEV, and VEEV. Recent advances in vaccine technologies have generated several new approaches to vaccine formulations that might be successfully applied to the development of new alphavirus vaccines [9] such as a multi-agent WEVEE virus-like replicon particle (WEVEE VRP).

The WEVEE VRP vaccine platform incorporates the packaging of a VEEV genome in which the structural proteins of VEEV are deleted and replaced with the foreign gene(s) of interest [10]. For the multi-agent alphavirus vaccine, the structural protein genes are replaced with the E1 and PE2 glycoproteins of VEEV, EEEV, or WEEV (WEVEE) [9,11]. The replicon RNAs lack the capsid protein and thus cannot undergo more than a single round of replication yet will express large amounts of the genes of interest once inside a mammalian cell. The WEVEE VRP vaccine is produced by electroporation of the replicon RNA and two helper RNAs that provide the structural proteins necessary for packaging RNA into a WEVEE VRP.

In a previous study, the WEVEE VRP vaccine protected mice from lethal subcutaneous and aerosolized VEEV, EEEV, and WEEV infection, and was effective against aerosolized VEEV and EEEV infection in cynomolgus macaques [9]. The previous cynomolgus macaque study demonstrated a trend toward protection against aerosolized WEEV; however, statistical significance was not achieved. Therefore, additional studies were necessary to determine if this vaccine is sufficiently effective to proceed with advanced development.

The objectives of this study were to examine the effect of the route of administration (intradermal, intramuscular, or subcutaneous) of the WEVEE VRP vaccine on immunogenicity against WEEV, EEEV, and VEEV and the efficacy of the combined vaccine against aerosolized WEEV in the cynomolgus macaque model of infection.

## 2. Materials and Methods

### 2.1. Study Materials

#### 2.1.1. WEVEE VRPs

The multi-agent WEVEE VRP vaccine material used in these studies was prepared and characterized by Alphavax (Research Triangle Park, NC, USA). The WEEV (pERK 247 EnG WEE 2100 VRP Lot 7/22/2010), EEEV (pERK 342 EnG EEE 4200 VRP Lot 8/11/2010), and VEEV (pERK 342 EnG VEE 3022 VRP Lot 8/11/2010) VRPs were purchased from AlphaVax.

For all routes of inoculation, primary and secondary vaccinations were prepared in the same manner. The starting concentrations of VEE, EEE, and WEE VRPs were 1.4 × 10^10^ infectious units (IU), 1.37 × 10^10^ IU, and 5.62 × 10^9^ IU, respectively. To inoculate seven NHPS by each route of inoculation, the volume of VRP needed to make a 1:1:1 ratio of 1.0 × 10^9^ IU for a total of 3.0 × 10^9^ IU WEVEE VRP was 0.500 mL (VEE VRP), 0.511 mL (EEE VRP) and 1.246 mL (WEE VRP).

For the intramuscular (IM) and subcutaneous (SC) inoculations, 11.75 mL sterile PBS (Corning, Corning, NY, USA) was added to the WEVEE VRP and the 14 mL mixture drawn up into seven 3 mL syringes (Becton-Dickinson, Franklin Lakes, NJ, USA) and placed on ice until used for IM injections into the caudal thigh or SC injection into the space between the scapula on each animal.

For the intradermal (ID) inoculations, seven 1 mL syringes (Terumo; Shibuya City, Tokyo, Japan) were filled with 0.3 mL of undiluted WEVEE VRP. The syringes were placed on ice until used for three ID injections (0.1 mL each) into the abdomen of each animal just above the umbilicus.

Control syringes were prepared by drawing 2.5 mL sterile PBS (Corning) into 3 mL (Becton-Dickinson) syringes for IM inoculation or 0.1 mL sterile PBS into 1 mL (Terumo) syringes for ID inoculation.

#### 2.1.2. Challenge Strain

The WEEV CBA87 challenge strain (USAMRIID) was originally isolated from an infected horse brain during an epizootic outbreak in 1958 (Cordoba, Argentina). The virus stock has a passage history of SM14, BHK2, V2 with the last two passages occurring in ATCC African Green Kidney Cells (Vero) cells. The cells were cultured in Minimum Essential Medium with Earl’s Salts (EMEM) (Corning) supplemented with 5% fetal bovine serum (Hyclone Laboratories, Logan, UT, USA), 10 mM HEPES (Sigma-Aldrich, St. Louis, MO, USA), 20 mM L-Glutamine (Hyclone Laboratories), and 1000 IU/mL and 1000 ug/mL Pen/Strep, respectively (Cellgro Technologies, Lincoln, NE, USA), and 50 ug/mL Gentamicin (Sigma-Aldrich). At approximately 32 h following inoculation, supernatants were harvested by centrifugation and precipitated overnight with 7% polyethylene glycol (Sigma-Aldrich) and 2.3% sodium chloride (Sigma-Aldrich). Next day, virus was pelleted by centrifugation (10,000× *g* for 30 min) and suspended in 6 mL 1× TNE. Continuous 20% to 60% sucrose gradients were prepared, samples were loaded onto gradients, and virus was purified by centrifugation (100,000× *g* for 3.5 h). Purified virus was harvested in 1 mL fractions from the virus band then assessed for protein concentration. Fractions with protein concentration ≥700 ug/mL were combined and stored at 4 °C until virus titer was determined. The pooled fraction was then aliquoted and stored at −60 to −90 °C.

### 2.2. Aerosol Generation

Animals were exposed to a target-presented dose of 3.0 × 10^7^ PFU WEEV CBA87 in the USAMRIID Head-Only Automated Bioaerosol Exposure System (ABES-II). The achieved aerosol challenge dose for each animal was calculated from the minute volume (MV) determined with a whole-body plethysmograph box using Buxco XA software (Data Sciences International, New Brighton, MN, USA). The total volume of aerosol inhaled was determined by the exposure time required to deliver the estimated inhaled dose. The aerosol challenge was generated using a 3-jet Collison Nebulizer (CH Technologies, Westwood, NJ, USA) to produce a highly respirable aerosol (1–3 µm particles at a flow rate of 7.5 ± 0.2 L/min).

Samples of the aerosol collected from the exposure chamber using an all-glass impinger (AGI) during each challenge were agarose plaque-titrated to determine the inhaled PFU for each animal. Actual inhaled dose was calculated as follows:Inhaled Dose (PFU) = Aerosol Concentration × Minute Volume × Run Time
where aerosol concentration is calculated as the amount of virus (PFU/mL) × volume in AGI sampler (mL)/sampler flow rate (mL/min) × sampling duration (min). Minute volume (mL/min); and run time (min) are the total time of exposure and air wash, respectively.

### 2.3. NHP Husbandry and Testing

Thirty-three healthy adult male cynomolgus macaques were acquired from the National Institutes of Health Animal Center (Poolesville, MD, USA). The mean age of the NHPs was 5.9 years (range: 5.3–8.7 years) and their mean weight was 5.5 kg (range: 4.1–7.4 kg). Each animal received a physical examination upon entrance to the USAMRIID facility and blood was collected to confirm no pre-existing immunity to alphaviruses as determined by plaque reduction neutralization titer (PRNT). No pre-existing immunity was detected.

NHPs were housed singly with physical enrichment (toys) and with visibility of other NHPs on the study protocol. NHP rooms were maintained on a 12-h light/dark cycle. Temperature in the animal rooms was maintained between 64 and 84 °F (18 and 29 °C) with humidity maintained at 30 to 70%. NHPs were fed a standard NHP diet consisting of primate chow supplemented with dietary enrichment (fruits and vegetables) at least three times per week. Approximately 21 days prior to vaccination, a Data Sciences International TA10TA-D70 telemetry device was surgically implanted into each NHP to capture temperature and activity data.

Animals were randomized into five groups consisting of either six or seven animals each. Three groups of seven animals each were vaccinated via the intradermal (ID), intramuscular (IM), or subcutaneous (SC) route with the WEVEE VRP vaccine (3.0 × 10^9^ infectious units (IU) total; 1.0 × 10^9^ IU each VEEV, EEEV, and WEEV VRPs). Two groups of six animals each received a placebo (phosphate buffered saline; PBS) ID or IM. A second round of vaccinations or placebo was given 28 days later. On days 28, 40, and 49 after primary vaccination, blood was collected to measure virus-specific neutralizing antibody responses.

On day 40 after primary vaccination, NHPs were transferred into the animal biosafety level-3 (ABSL-3) containment facility for acclimation and collection of baseline telemetry data. After acclimation, the NHPs were challenged by aerosol with a target dose of 3.0 × 10^7^ PFU of WEEV CBA87 using a USAMRIID-designed head-only exposure system. Blood was collected on each of three days prior to challenge to establish baseline hematology, clinical chemistry, and cytokine levels for each NHP. Due to the size of this study, the challenge portion was conducted over two days; ID-vaccinated and ID-control animals were aerosol-exposed on day 56 after primary vaccination; IM-vaccinated, IM-control, and SC-vaccinated animals were aerosol-exposed on day 57 after primary vaccination. The challenge agent was prepared on day 56 for all sprays. Unused agent was frozen at −60 to −90 °C and thawed on ice for use the following day. Following exposure to WEEV CBA87, NHPs were observed for clinical signs of disease at least twice daily. Daily clinical observation scores were a composite of several key parameters including temperature, neurological signs (e.g., tremors, seizures), physical appearance, as well as natural and provoked behavior. For all control and WEVEE VRP-vaccinated NHPs, a daily clinical observation score of 5 or greater prompted a third daily observation of the affected NHP, and a daily score of 12 or greater signaled that the NHP met euthanasia criteria. Blood was collected on days 1 to 10, 12, 14, 16, 18, 21, 24, and 28 post exposure (PE), and at time of euthanasia for hematology and chemistry. At time of euthanasia, tissues were collected for plaque assay and real-time PCR. At study conclusion or when moribund, NHPs were humanly euthanized in accordance with institute protocols and necropsied.

### 2.4. Plaque Reduction Neutralization Titer (PRNT) Assay

The PRNT assay was used to assess neutralizing antibody levels present in serum. Briefly, serum samples were heat-inactivated for 30 ± 1 min at 56 ± 2 °C. Samples were diluted 1:20 in Hank’s Balanced Salt Solution (HBSS) (Cellgro), supplemented with 2% heat-inactivated FBS (Hyclone), 2% Pen/Strep (Cellgro), and 1% HEPES (Sigma-Aldrich), and then serially diluted 1:2. Virus stocks were diluted to a concentration of 2.0 × 10^3^ pfu/mL (2× virus stock) and added 1:1 to the serially-diluted samples. Samples were incubated overnight at 2–8 °C. The 2× virus stock was diluted 1:1 in HBSS, 2% heat-inactivated FBS and 1% HEPES and was used to determine the plaque count where 80% neutralization occurred (PRNT80). This virus stock sample was also incubated overnight at 2–8 °C. USAMRIID Vero 76 cells seeded on 6-well plates were grown to approximately 90–100% confluence. Cells were infected with 0.1 mL of each serial dilution per well in duplicate. Plates were incubated at 37 ± 2 °C for 1 h ± 15 min with gentle rocking every 15 min. After 1 h, cells were overlaid with agarose 0.6% in Basal Medium Eagle (BME; Thermo Fisher Scientific (Gibco) Waltham, MA, USA) supplemented with 10% heat-inactivated FBS, and 2% Pen/Strep. Plates were incubated 24 ± 4 h at 37 ± 2 °C, 5 ± 1% CO_2_. A second overlay containing 0.6% agarose in BME supplemented with 10% heat-inactivated FBS, 2% Pen/Strep, and 5% neutral red vital stain (Thermo Fisher Scientific (Gibco)) was added to wells and further incubated 18–24 h for visualization of plaques and determination of PRNT80 value.

### 2.5. Enzyme-Linked Immunosorbent Assay (ELISA)

The ELISA assay was used to assess the binding IgG levels present in serum. Briefly, 96-well ELISA plates were coated overnight at 2–8 °C with 50 µL of sucrose purified virus in PBS (concentration of 6 µg/mL for VEEV and WEEV, and 4 µg/mL for EEEV). Following incubation, excess was decanted into a pan of disinfectant and 300 µL PBS with 0.02% Tween-20 and 5% nonfat dry milk (PBSTM) was added to each well as a blocking agent and incubated for 2 h at 37 °C. Following incubation, plates were either used immediately or stored at −20 °C for use on a later date. After blocking, plates were washed three times with 300 µL of PBS with 0.02% Tween-20 (PBST). Samples were added to the plate in duplicate and diluted down the plate in two-fold dilutions in PBSTM and 1% heat-inactivated FBS (HI-FBS) starting at a dilution of at least 1:40 at final volumes of 50 µL. Plates were then incubated for 1–2 h at 37 °C followed by washing with 300 µL of PBST three times. After washing, 100 µL of goat anti-human IgG horseradish peroxidase-conjugated antibody (KPL, Gaithersburg, MD, USA) diluted 1:50,000 for EEEV and VEEV and 1:100,000 for WEEV in PBSTM plus 1% HI-FBS was added and then incubated for 1 h at 37 °C. After another three washes with 300 µL PBST, 100 µL of ABTS substrate (KPL) was added and incubated approximately 10 min at 37 °C. Plates were read at 410 nm on a SpectraMax M5 (Molecular Devices, San Jose, CA, USA).

### 2.6. Virus Quantitation

#### 2.6.1. Plaque Assay

Whole blood was collected and aliquoted for storage at −60 to −90 °C for future analysis. Infectious virus in blood was quantitated by plaque assay [10]. Briefly, serum was ten-fold serially diluted starting at a dilution of 1:10 in Eagle’s Minimum Essential Medium (EMEM) supplemented with non-essential amino acids (NEAA; Sigma M7145), 2% FBS (Hyclone SH30071.03), 2% Pen/Strep (Cellgro 30-002-CI), and 1% HEPES (Sigma, H0887). USAMRIID Vero 76 cells were seeded on 6-well plates to be approximately 90–100% confluent at use. Cells were infected with 0.1 mL of each dilution per well in duplicate. Plates were incubated at 37 °C for 1 h with gentle rocking every 15 min. After 1 h, cells were overlaid with 0.6% agarose in BME (Gibco A15950DK) supplemented with 10% FBS, 2% Pen/Strep, 1% NEAA, 1% L-glutamine (Hyclone SH30034.01), and 1% gentamicin (Sigma G1397). Plates were incubated 24 h at 37 °C, 5% CO_2_. A second overlay containing 0.6% agarose in BME supplemented with 10% FBS, 2% Pen/Strep, 1% NEAA, 1% L-glutamine, 1% gentamicin, and 5% neutral red vital stain (Thermo Fisher Scientific (Gibco)) was added to wells and further incubated 18–24 h for visualization of plaques.

#### 2.6.2. Real-Time PCR

To corroborate results of the plaque assays, semi-quantitative real-time polymerase chain reaction (RT-PCR) was utilized to measure the number of WEEV CBA87 viral genome copies per milliliter of blood, cerebral spinal fluid (CSF), or per milligram of tissue. Viral RNA was extracted from blood, CSF, and tissues using commercially available kits (QIAmp Viral RNA kits, Qiagen) according to manufacturer’s instructions.

The USAMRID-designed WEEV viral RNA assay amplified a 70 bp portion of a structural polyprotein of the CBA87 strain of WEEV. The assay limits of detection ranged from 5.0 × 10^7^ synthetic RNA copies/µL (upper limit of detection [ULOD]) to 5.0 × 10^2^ synthetic RNA copies/µL (lower limit of detection [LLOD]). Positive and negative extraction controls (PEC and NEC, respectively) were created by supplementing uninfected control NHP blood with a known amount of WEEV CBA87 virus (1.0 × 10^4^ PFU for the PEC), or RNAse-free water (NEC). Positive and negative template controls (PTC and NTC, respectively) were created by supplementing uninfected control NHP blood with a known amount of synthetic RNA (1.0 × 10^4^ genome equivalents (ge)/mL for the PTC), or RNAse-free water (NTC). This assay recognizes the structural protein which will detect both genomic and subgenomic copies of viral RNA. Although not ideal for a quantitative assay, the technique does provide a qualitative assay and increased sensitivity for detection. The results were reported as the absence or presence of viral RNA.

### 2.7. Clinical Pathology

Blood samples were collected on three days prior to WEEV CBA87 exposure and on days 1–10, 12, 14, 16, 18, 21, 24, 28 PE, and at time of death for hematology and serum chemistry analysis.

#### 2.7.1. Hematology

Whole blood was collected into pediatric blood tubes containing K2 EDTA as an anti-coagulant. Tubes were gently inverted by hand and/or on a tube rocker to ensure adequate mixing and kept at ambient temperature. A quantitative blood cell count was conducted using a Cell Dyn 3700 clinical hematology analyzer (Abbott, Abbott Park, IL, USA) within 2 h of collection in order to obtain accurate counts. Normal values for cynomolgus macaques were determined by the USAMRIID Clinical Laboratory Department, Division of Medicine, based on historical hematology data from the USAMRIID cynomolgus macaque colony.

#### 2.7.2. Serum Chemistry

Whole blood was collected into serum separator tubes, which were gently inverted by hand to ensure adequate mixing, and placed upright at ambient temperature. Tubes were allowed to clot for approximately 30 min and the serum separated in a centrifuge set at 1500× *g* for 10 min at ambient temperature. The required volume of serum was removed for chemistry analysis using a Vitros 250 clinical chemistry analyzer (Ortho Clinical Diagnostics, Raritan, NJ, USA). Serum was removed from the separator tubes within 15 min of centrifugation and was analyzed within 48 h of collection. Normal values for cynomolgus macaques were determined by the USAMRIID Clinical Laboratory Department, Division of Medicine, based on serum chemistry data from the USAMRIID cynomolgus macaque colony.

### 2.8. Pathology

A full necropsy of each animal was performed in BSL-3. Tissues were collected for virus isolation, histology, and immunohistochemistry. For virus isolation by plaque assay and RT-PCR, the following tissues were collected: axillary lymph node, inguinal lymph node, mandibular lymph node, mandibular salivary gland, tonsil, tongue, tracheobronchial lymph node, lung, heart, liver, spleen, adrenal gland, pancreas, kidney, mesenteric lymph node, cecum, gonad (testis), urine, bone marrow, brain (cerebrum and cerebellum), olfactory bulb, cervical spinal cord, eye, popliteal lymph node, and cerebral spinal fluid. Each of these samples was collected aseptically and put into individually-labeled cryopreservation tubes and flash frozen on dry ice.

For histology and immunohistochemistry, the following tissue were collected: brain (frontal cortex, corpus striatum, thalamus, mesencephalon, pons, medulla oblongata, hippocampus, cerebellum, and olfactory bulb), cervical spinal cord, lung, spleen, mesenteric lymph node, tracheobronchial lymph node, axillary lymph node, popliteal lymph node, tonsil, bone marrow, nasal cavity (olfactory epithelium), liver, pancreas, adrenal gland, and testis.

#### 2.8.1. Histology

The histology tissue samples from each NHP were fixed by immersion in containers of 10% neutral buffered formalin labeled with the animal’s identification number. Each container of tissues was held for a minimum of 21 days under BSL-3 containment and then was surface-decontaminated and transferred from the biocontainment suite to the USAMRIID histopathology laboratory. After arrival at the histopathology laboratory, the tissue samples were trimmed, routinely processed, and embedded in paraffin. Sections of the paraffin-embedded tissues 5 µm thick were cut for histology. The histology slides were deparaffinized, stained with hematoxylin and eosin (H&E), cover slipped, and labeled.

#### 2.8.2. Immunohistochemistry

Replicate sets of the histology slides were made for IHC. Serial sections of select tissues were cut and stained for viral antigen using a polyclonal rabbit antiserum directed against several alphaviruses, followed by a horseradish peroxidase-labeled polymer conjugated to goat anti-rabbit immunoglobulins.

Briefly, tissue sections were deparaffinized using Xyless (LabChem, Inc., Pittsburgh, PA, USA) and rehydrated using sequentially less concentrated alcohol solutions ranging from 100% to 70%. Endogenous peroxidases were blocked using a methanol/hydrogen peroxide solution. To increase staining intensity, antigen retrieval was performed by immersing slides in Tris/EDTA buffer for 30 min at 97 °C. Endogenous proteins were blocked by incubating the slides in serum-free protein block (Invitrogen, Carlsbad, CA, USA) supplemented with 5% normal goat serum (Vector Labs, Burlingame, CA, USA) for 30 min at room temperature. Sections were incubated with the primary antibody, a polyclonal rabbit antiserum directed against EEEV, WEEV, VEEV, or Sindbis virus (Applied Diagnostic Branch, Diagnostic Systems Division, USAMRIID, immune #1140) diluted 1:8000, for 30 min at room temperature. Sections were then incubated with a secondary antibody, horseradish peroxidase-labeled polymer conjugated to goat anti-rabbit immunoglobulins (DAKO, Carpenteria, CA, USA), and incubated for 30 min at room temperature. Staining was completed by adding the substrate-chromagen, diaminobenzidine (DAB) (DAKO) and incubating slides for 5 min at room temperature. Tissues were counterstained with hematoxylin for 2 min at room temperature and then dehydrated in sequentially more concentrated alcohol solutions, cleared using Xyless II, and coverslip was mounted using Permount (Fisher Scientific, Pittsburgh, PA, USA). Non-immune (normal) rabbit serum (Vector Labs) was used as a negative control for the primary antibody. Sections of confirmed WEEV infected cynomolgus brain were used as positive controls.

All tissues that were collected and processed for H&E were also evaluated for distribution of staining. The tissue sections (grey matter only in the brain sections) were graded on the following scale to classify the percentage of cells that exhibited positive immunohistochemical staining for alphavirus antigen: 0 = Zero, no cells in section are positive (negative); 1 = 10% or fewer of cells in section are positive (minimal); 2 = 11–25% of cells in section are positive (mild); 3 = 26–50% of cells in section are positive (moderate); 4 = 50–75% of cells in section are positive (marked); 5 = more than 75% of cells in section are positive (severe). Only those tissues with positive staining were recorded.

### 2.9. Statistical and Temperature Analysis

Individual NHP temperatures were modeled using R statistical software (R Foundation for Statistical Computing, Vienna, Austria) and the remaining statistical analyses were conducted using SAS Version 9.2 (SAS Institute, Cary, NC, USA). All tests were two-sided (indicated treatment differences whether positive or negative) whenever the probability of a difference occurring by chance was less than 0.05 (alpha). Differences between treatment groups were analyzed using Fisher’s Exact Test, log-rank tests associated with Kaplan-Meier curves, exact Wilcoxon tests. Locally Weighted Polynomial Regression (LOESS) was used to visualize measurement results other than temperature. Summary statistics were calculated for each treatment group.

Methods for handling missing data and outliers were utilized as follows for analog temperature data. A number of different methods were used to describe and identify temperature measurements as missing data or potential outliers. Outliers were defined as measurement results that may not be rational from a physiological perspective (non-physiological). Missing on Delivery: These data are simply missing. This category of data was included when summarizing missing data or outliers. Model Rejected: If the software for determining high temperatures failed to provide any model for temperatures, a set of data from the beginning of the intended baseline period were set to missing until the software did produce a model. High: Measurements were set to missing if they were above 41 °C. Unlikely Jump: Both the measurement before and after a temperature change greater than 2 °C in a 15 min interval were set to missing. Missing Comparators: Measurements were set to missing if they were not associated with sufficient data to calculate quality control limits. Above or below QC: Measurements higher/lower than the upper/lower limit for quality control were set to missing. Quality control limits were calculated as the 95% prediction interval for a single measurement. The associated moving average was calculated using the current temperature value, temperature values from the previous two time periods (15 min each) and temperature values from the following two time periods. Accepted: Accepted data were used as measured. They consisted of data not rejected due to any of the preceding criteria.

## 3. Results

### 3.1. Antibody Responses, PRNT and ELISA

Twenty-one cynomolgus macaques (three groups, *n* = 7 NHP/group) received WEVEE VRP intramuscularly (IM), intradermally (ID), or subcutaneously (SC). Twelve cynomolgus macaques (two control groups, *n* = 6 NHP/group) received PBS IM or ID. Whole blood was collected from NHPs and serum was isolated to assess the levels of binding and neutralizing antibodies present. Neutralizing antibody titers were measured by PRNT assay on sera collected prior to and on days 28, 40, and 49 after primary vaccination. Total binding IgG levels were measured by ELISA using sera collected prior to and on days 28 and 49 after primary vaccination. All pre-vaccination PRNT and ELISA levels were negative (<1:20) for all animals and all PBS-inoculated controls animals remained negative through day 49. Based on PRNT assay results, WEVEE VRP vaccination resulted in the production of VEEV, EEEV, and WEEV neutralizing antibody responses that were enhanced by homologous boost vaccination on day 28 after primary vaccination (Figure 1). Comparable levels of neutralizing antibody responses were stimulated when NHPs were vaccinated ID, IM, or SC. Peak neutralizing titers were reached by day 40 after primary vaccination and neutralizing titer levels were largely sustained through day 49. Based on ELISA, WEVEE VRP vaccination resulted in the production of VEEV, EEEV, and WEEV binding antibody responses that were enhanced by homologous boost vaccination on day 28 after primary vaccination (Figure 2). Comparable levels of binding antibodies were stimulated when NHPs were vaccinated ID, IM, or SC.

### 3.2. WEEV CBA87 Aerosol Exposure

Cynomolgus macaques (*n* = 33) were aerosol-exposed to a target-presented dose of 3.16 × 10^7^ PFU of WEEV CBA87. All animals received similar doses of aerosolized WEEV CBA87 with calculated inhaled doses ranging from 5.51 × 10^6^ to 4.13 × 10^7^ PFU and averaging at 1.41 × 10^7^ PFU. The calculated average doses for the ID- and IM-control groups were 1.68 × 10^7^ and 1.43 × 10^7^ PFU, respectively. The WEVEE VRP-vaccinated groups’ calculated average doses were 1.47 × 10^7^, 1.27 × 10^7^, and 1.26 × 10^7^ PFU for the ID, IM, and SC routes, respectively.

To confirm WEEV exposure, NHP nasal and throat swab samples were collected immediately following virus exposure. The presence of infectious virus was determined by plaque assay. Virus was detected from both nasal and throat swabs collected from each NHP confirming that every NHP received aerosol exposure to WEEV CBA87.

### 3.3. WEEV CBA87 Mortality and Morbidity

#### 3.3.1. Mortality

One ID-control NHP, ID CTRL-1, met euthanasia criteria on day 8 PE with a clinical score of 20 and was humanely euthanized. A second ID-control NHP, ID CTRL-6, was removed from study due to complications and loss of a functional telemetry device. This NHP was not included in any additional analyses reported. All IM-control and all WEVEE VRP-vaccinated NHPs survived to study end-point (Figure 3). Examining IM- (*n* = 6) and ID- (*n* = 5) control animals together, WEEV aerosol challenge resulted in 9% mortality overall. There were no statistically significant differences in survival rates between the WEVEE VRP-vaccinated NHPs and the control NHPs.

#### 3.3.2. Morbidity

In the absence of a lethal disease model, other factors must be analyzed to determine the efficacy of the WEVEE VRP vaccine. One such measure is a decrease in the overall clinical scores of the vaccinated NHPs in comparison to their respective control groups.

The average clinical scores of control animals were higher than vaccinated animals between days 5 and 10 PE (Figure 4). Only one of the five ID-control animals met euthanasia criteria (ID CTRL-1). Two ID-control NHPs (IDCTRL-1, and -5) and four-IM control NHPs (IM CTRL-1, 3, 4, and 5) displayed clinical signs of disease resulting in scoring ≥5 between days 5 and 11 PE, and only one WEVEE VRP-vaccinated NHP from each group (ID VRP-2, IM VRP-6, and SC VRP-5) displayed clinical disease resulting in scoring ≥5. IM VRP-6 had a clinical score of 6 on day 9 PE. The other two vaccinated animals displayed elevated clinical scores beyond day 11 PE; ID VRP-2 had a clinical score ≥ 5 on days 16 and 17 PE and SC VRP-5 had a clinical score of 5 on day 23 PE.

A comparison of the average clinical scores from the WEVEE-vaccinated NHPs shows a similar pattern of clinical manifestations regardless of the route of vaccination (ID, IM, or SC; Figure 5A). Analysis of the individual area under the curves (AUC) per NHP from each treatment group (Figure 5B) reveals a trend, though the differences did not reach statistical significance, of lower AUC values after vaccination in comparison to the control NHPs.

To examine the ability of the WEVEE VRP-vaccinations to reduce the presence of potential illness or neurological signs, the number of days on which tremors were observed were recorded. The average number of days a WEEV-exposed NHP had an observed tremor was reduced in WEVEE VRP-vaccinated NHPs. On average, NHPs that received the IM WEVEE VRP-vaccination had 2.3 days of tremors, NHPs that received the ID WEVEE VRP-vaccination had 2.4 days, and NHPs that received the SC WEVEE VRP-vaccination had 4.7 days of tremors following WEEV-exposure. In comparison, IM- and ID- control NHPs averaged 6.2 and 7.6 days of tremors, respectively. Based on this assessment, VRP-vaccination appeared to reduce the number of days of observed potential neurological symptoms with the hierarchy of tremor days experienced being IM- < ID- < SC- < control NHPs.

### 3.4. Temperature Monitoring after WEEV CBA87 Exposure

Temperature data collected for at least 7 days prior to aerosol exposure were used to establish a baseline or normal average of individual NHP temperatures for each hour of the day. Standard deviation (SD) was calculated and fever threshold was set for each hour as 3 SDs greater than the average temperature at each specific hour. Temperatures falling above the fever threshold were labelled *High* and used to determine the total fever hours for each NHP (Figure 6). Potential temperature outliers were removed based on definitions outlined in the methods section. On average, both ID- and IM-control groups had greater total hours of elevated temperatures compared to all three WEVEE VRP-vaccinated groups which had little to no fever measured. The differences in median total fever hours between control and vaccinated NHPs were statistically significant (*p* < 0.04).

### 3.5. WEEV CBA87 Virus Quantification in Blood and Tissues

Corroborating observations from other WEEV NHP studies [3,9,12], viremia was not detected in the blood of any of the NHPs by plaque assay between days 1 to 7 following aerosol exposure to WEEV CBA87. The real-time PCR assay that was utilized for this study was qualitative in nature, as it only included a semi-quantitative RNA standard. For this reason, Table 1 reports the presence or absence of viral genome amplification in blood samples. Amplification of signal above the LLOD is reported as positive. Viral genomic material was detected in the blood of control animals between days 1 and 3 PE. Positive identification of viral genomes in blood occurred in 5 of 11 control NHPs. Consistent amplification below the LLOD occurred in 2 NHPs; however, some control NHPs had no detectable levels of viral RNA (ID CTRL-4, IM CTRL-1, and IM CTRL-3).

The dissemination of virus infection in the NHPs at the time of euthanasia following exposure to WEEV CBA87 was assessed by plaque assay and RT-PCR analysis of tissue homogenates. A total of 26 tissues were collected from each NHP; however, only a subset of relevant tissues was assessed for the presence of infectious virus or viral genomes. Cerebellum, cerebrum, brainstem/spinal cord, mandibular lymph node, spleen, liver, lung, tonsil tissue, as well as cerebral spinal fluid (CSF) from the ID-control NHP that met euthanasia criteria (ID CTRL-1) were tested for the presence of infectious virus by plaque assay. Infectious virus was not detected in any of the tissues assayed. There was insufficient olfactory bulb sample to evaluate the presence or absence of infectious virus by plaque assay.

Table 2 summarizes the RT-PCR results in tissues and CSF. Amplifiable viral genome copies were absent from all vaccinated NHPs at the end of study. ID CTRL-1, which met euthanasia criteria on day 8 PE, had detectable viral RNA in central nervous system-associated tissues including the olfactory bulb, cerebellum, brainstem/spinal cord, and CSF. Other control NHPs with detectable RNA in the olfactory bulb at end of study (≥28 days PE) included ID CTRL-4, ID CTRL-5, and IM CTRL-1. Additionally, ID CTRL-5 had amplifiable RNA levels (1 of 2 independent PCR runs) in the brainstem/spinal cord.

### 3.6. Clinical Pathology after WEEV CBA87 Exposure

Hematology and serum chemistry data were assessed for the animals in this study starting three days prior to challenge and on days 1–10, 12, 14, 16, 18, 21, 24, and 27/28 PE. For ID CTRL-1 samples were collected at the prescribed schedule until time of euthanasia on day 8 PE. Overall, there were no statistically significant differences in the hematology and serum chemistry results between the control groups and the vaccinated groups. Multiple animals, representing all groups, exhibited a leukocytosis, most commonly characterized by a monocytosis and/or neutrophilia, and often accompanied by a lymphopenia (Figure 7). The minimal to mild leukocyte elevations tended to occur during the first two weeks after exposure to WEEV, implicating viral exposure as the most likely contributing factor. It may be worth noting that the animals in the control groups exhibited the overall longest spanning frequencies of leukocytosis, although there was wide variability in occurrences and frequencies overall.

### 3.7. Pathology

A full necropsy was performed on all NHPs including the two animals that were euthanized on day 8 PE (ID CTRL-6 was euthanized for reasons unrelated to WEEV disease). Tissue was collected for virus isolation, histology, and immunohistochemistry. Gross findings referable to the CNS were observed in two control NHPs. ID CTRL-6 had slight meningeal congestion. ID CTRL-5 had caudal cerebral hemispheres dilated bilaterally with clear fluid in the lateral ventricles, with attenuated cerebral cortex (hydrocephalus) and mildly swollen cerebral gyri and vascular congestion. The two control NHPs euthanized on day 8 PE had detectable viral antigen within the brain by immunohistochemistry (Figure 8A); no detectable viral antigen was present in tissues from the 10 animals that were euthanized at the end of study (Figure 8B). Eleven of twelve (92%) control NHPs had histopathological findings of encephalitis or meningoencephalitis consistent with WEEV infection in the CNS at euthanasia or time of death (Figure 8E,F); one control NHP had no significant lesions in the CNS (Appendix A). Six of twenty-one (33%) of the vaccinated NHPs had perivascular cuffs (perivascular mononuclear cell infiltrates) in the gray matter, white matter or meninges. It is unknown whether these very minor changes in the WEVEE VRP-vaccinated NHPs were related to the virus exposure or were incidental findings. Perivascular inflammatory cell infiltrates have been reported to occur rarely as incidental findings in cynomolgus macaques [13,14]. Lymphoid organs (spleen, tonsils, or lymph nodes) of all control and vaccinated macaques had lymphoid hyperplasia, attributed to immune stimulation induced by aerosol exposure to the virus. Viral antigen was not detected in any tissue of any vaccinated NHP.

### 3.8. WEEV CBA87 Post-Exposure Neutralization Titers

Blood was collected at the time of euthanasia (day 8 PE) for ID CTRL-1 and on day 28 PE for all other NHPs to measure the level of WEEV-specific neutralizing antibodies present after aerosol exposure to WEEV CBA87. Figure 9 shows the PRNT80 values at the time of euthanasia or on day 28 PE for each NHP. As expected, all WEVEE VRP-vaccinated NHPs had high PRNT80 values. The control NHP that met euthanasia criteria (ID CTRL-1) had a measurable neutralizing titer at day 8 PE. Control NHPs (ID CTRL-2 to -5 and IM CTRL-1 to -6) that survived to study end-point (≥day 28 PE), were seropositive on day 28 PE with low-level PRNT80 values. These data demonstrate that the NHPs were productively infected and mounted an immune response.

## 4. Discussion

The purpose of the study reported here was to expand upon a previous investigation of the trivalent WEVEE VRP vaccination in nonhuman primates that demonstrated vaccination-elicited strong neutralizing antibody responses against VEEV, EEEV, and WEEV, albeit lower responses against WEEV [9]. In this study, we examined the effect of route of administration (intradermal, intramuscular, or subcutaneous) of the WEVEE VRP vaccine on immunogenicity against WEEV, EEEV, and VEEV and efficacy against aerosolized WEEV in the cynomolgus macaque model of infection. We observed that, regardless of the route of administration, the WEVEE VRP vaccine was immunogenic and provided protection against WEEV aerosol exposure in NHPs.

Multiple routes of vaccination were evaluated in this study to determine if inoculation through a route that mimics a mosquito bite would provide comparable immune stimulation to the traditional IM vaccination. Until recently, IM vaccination was predominately used because it is easy to perform and well tolerated by patients. As knowledge about the immune competence of the skin expands [15,16] alternative routes of vaccination such as SC and ID have been explored [16]. We found that vaccination with two doses of WEVEE VRP (3.0 × 10^9^ IU total; 1.0 × 10^9^ IU each VEEV, EEEV, and WEEV VRPs) resulted in neutralizing and binding antibody responses specific for VEEV, EEEV, and WEEV regardless of the route of administration. Further, there were no statistical differences in the levels of neutralizing or binding antibodies stimulated after the different routes of vaccination. These data suggest that route of vaccination had no impact on the ability of the vaccine to stimulate a humoral response, and corroborates findings by others that SC and ID vaccination could be used interchangeably with IM vaccination [17,18,19,20,21].

Pittman et al. previously reported diminished neutralizing antibody responses in patients that received WEEV or EEEV vaccine prior to the live attenuated VEE TC-83 vaccine [22]. This finding, consistent with studies that demonstrated that patients previously vaccinated with a Chikungunya virus vaccine failed to develop anti-VEEV neutralizing antibodies upon VEE TC-83 vaccination [23], suggests that immune interference occurs after sequential alphavirus vaccinations. Here, NHPs were vaccinated concurrently with equal amounts of VEE, EEE, and WEE VRPs to avoid any potential immune interference. This strategy was demonstrated previously to mitigate the risk of sequential vaccination-related immune interference observed in earlier NHP studies [9,12,24]. We observed lower anti-WEEV neutralizing and binding antibody responses in comparison to the anti-VEEV and EEEV levels elicited after vaccination with the WEVEE VRP. However, the levels measured were comparable to those reported by Ko et al. after vaccination of NHPs with a WEVEE virus-like particle vaccine [12]. Further, a comparison of the levels of anti-WEEV neutralizing antibodies in WEVEE VRP-vaccinated NHPs prior to exposure (range of dilution, 1:80–1:640) and the control animals at the end of study (range of dilution, 1:40–1:320) finds similar levels of neutralizing antibody produced from vaccination as a natural infection. These data suggest that the WEEV antigen, regardless of the source, is less immunogenic than antigen from VEEV and EEEV.

Performance of medical countermeasure (MCM) efficacy studies under the FDA Animal Rule requires “the response to the challenge agent manifested by the animal species should be similar to the disease or condition in humans exposed to the etiologic agent” [25]. This is particularly challenging for model development of WEEV disease as there are few reports in the literature about WEEV disease in humans [26,27,28,29]. Development of an appropriate WEEV animal model with consistent and reproducible markers of infection is needed if MCM products are intended to be licensed under the FDA Animal Rule. In this study only one control animal out of 11 met euthanasia criteria due to WEEV disease on day 8 PE, resulting in a mortality rate of 9%, consistent with what has been observed in humans. However, the low mortality rate of WEEV in NHPs means other markers of infection and disease are required to determine MCM efficacy. Statistically significant differences in median fever hours were noted in control NHPs compared to vaccinated NHPs, providing a quantitative measure of infection and efficacy of the vaccine against a WEEV challenge. Additionally, the frequency of potential neurological signs (e.g., tremors) observed in the control and VRP-vaccinated NHPs appeared to be reduced in vaccinated NHPs compared to control NHPs with those vaccinated through the IM and ID route having fewer recorded tremors than those vaccinated by the SC route. However, this measurement is subjective and relies on an observer’s assessment rather than a directly quantitative measure. Alterations in lymphocytes, monocytes, and neutrophils were observed. Lymphopenia was observed in control NHPs (LOWESS analyses); however, the changes were not statistically significant when compared by group by day. Ultimately, more work is needed to identify and evaluate additional biomarkers of WEEV disease for the cynomolgus macaque model to be used for MCM licensure.

## Figures and Tables

**Figure 1 viruses-14-01502-f001:**
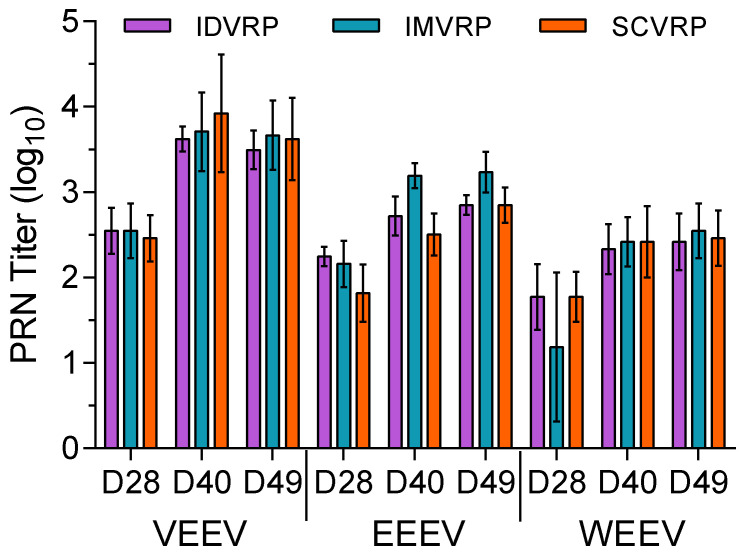
PRNT for sera collected from WEVEE VRP-vaccinated NHPs. Neutralization titers were measured from sera collected prior to and on days 28, 40, and 49 after primary vaccination. Neutralizing antibody levels against VEEV, EEEV, and WEEV were assessed. There were no detectable serum neutralizing antibodies in the pre-vaccination or CTRL (PBS) vaccinated NHP sera tested (*n* = 7 WEVEE VRP NHPs and *n* = 6 CTRL NHPs for each challenge virus). Error bars represent ± SD.

**Figure 2 viruses-14-01502-f002:**
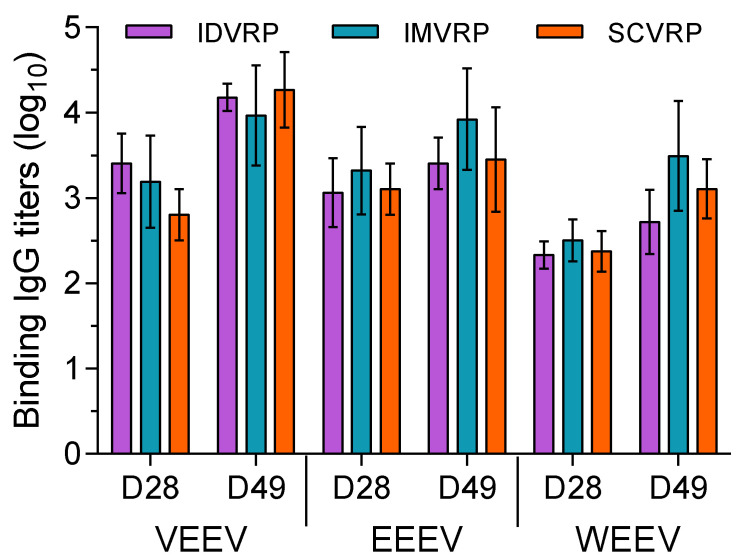
ELISA for sera collected from WEVEE VRP-vaccinated NHPS. Total serum IgG antibody titers were measured from sera collected prior to and on days 28 and 49 after primary vaccination. Binding IgG antibody levels against VEEV, EEEV, and WEEV were assessed. There were no detectable serum neutralizing antibodies in the pre-vaccination or CTRL (PBS) vaccinated NHP sera tested (*n* = 7 WEVEE VRP NHPs and *n* = 6 CTRL NHPs for each challenge virus). Error bars represent ± SD.

**Figure 3 viruses-14-01502-f003:**
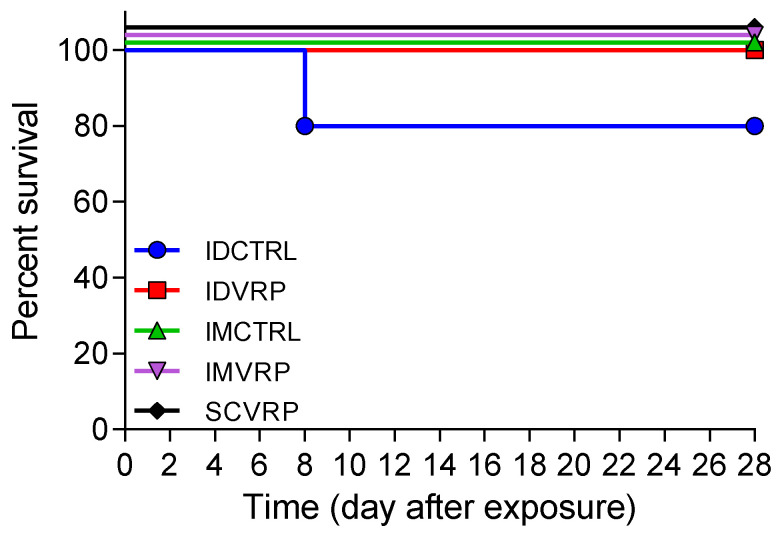
NHP percent survival after exposure to WEEV CBA87. One of five ID-control NHPs met euthanasia criteria on day 8 post exposure to WEEV CBA87 (80% survival rate). NHPs were euthanized when a clinical score of 12 (or greater was observed or at end of study. All other control and all WEVEE VRP-vaccinated NHPs survived to study end-point.

**Figure 4 viruses-14-01502-f004:**
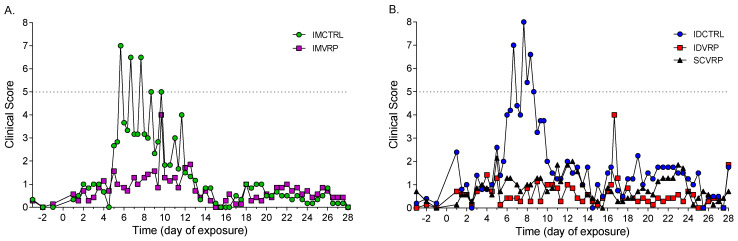
Average clinical scores. (**A**) IM-control and IM WEVEE VRP-NHPs (*n* = 6, and 7 respectively). (**B**) ID-control, ID WEVEE-VRP, and SC WEVEE VRP-NHPs (*n* = 5, 7, and 7 respectively).

**Figure 5 viruses-14-01502-f005:**
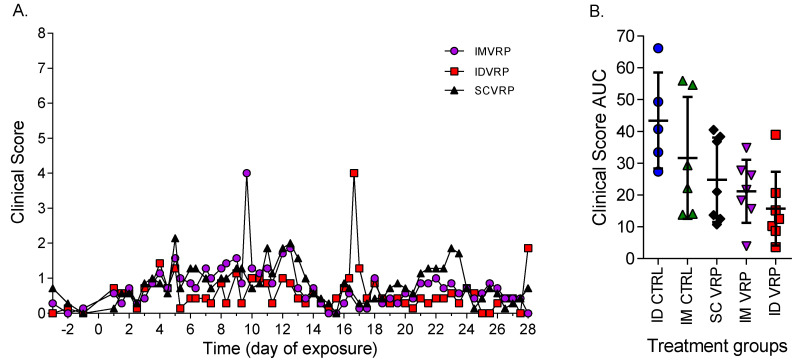
Clinical scores of WEVEE VRP-vaccinated NHPs. (**A**) Average clinical scores for ID, IM, and SC VRP-vaccinated NHPs (*n* = 7/group). All vaccinated NHPs survived to study end-point. The peak for the IM group on day 9 post exposure (PE) and the ID group on day 15 PE are due to one NHP with a score of 6 at each time point. (**B**) Area under the curves for each NHP per treatment group. Dots represent individual animals; bars represent the mean; error bars represent ± standard deviation.

**Figure 6 viruses-14-01502-f006:**
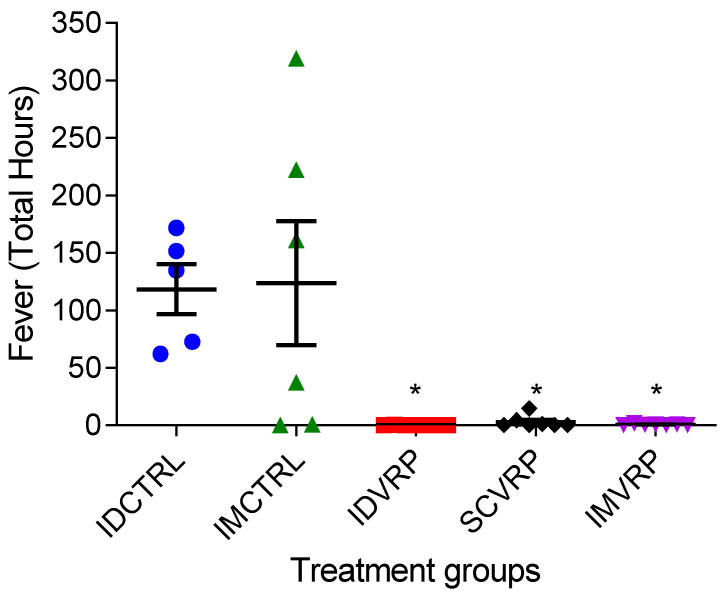
Total Fever Hours after aerosol exposure to WEEV CBA87. Any High temperature defined as Onset of Fever, End of Fever, or between Onset of Fever and End of Fever is defined as Fever. Any temperatures measured above the upper limit of the 99.7% highest posterior density credible interval of the error distribution was labeled as High. High temperatures are not automatically assumed to indicate fever. The total fever hours are the total time (in hours) associated with data that occur after exposure and are defined as Fever. Each symbol represents an individual NHP. The median difference in hours of fever per group (Median Fever Hours) was contrasted using the Wilcoxon Exact Test. ID VRP, SC VRP and IM VRP all have a lower median difference in fever hours than their associated controls with “statistically significant” test-wise *p*-values less than 0.05 (*).

**Figure 7 viruses-14-01502-f007:**
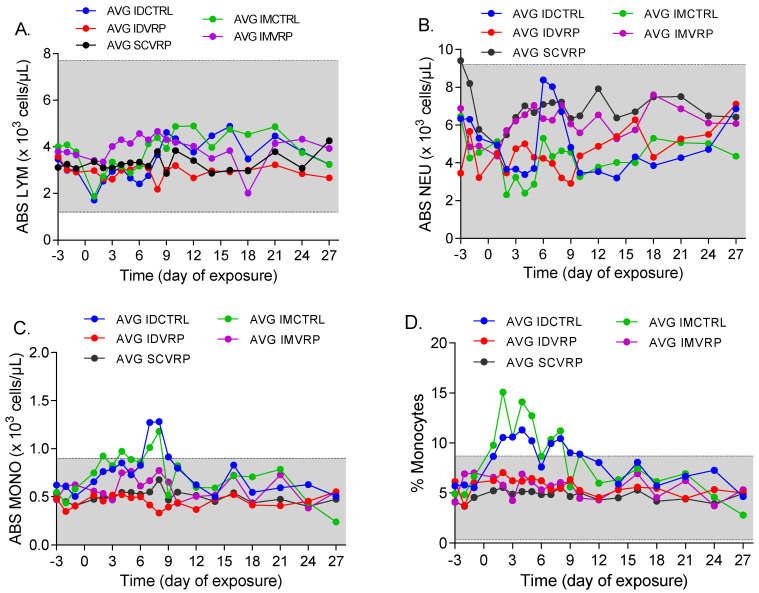
Hematology changes in NHPs following WEEV CBA87 exposure. Blood samples were collected on days −3 thru −1 before exposure and days 1 thru 10, 12, 14, 18, 21, 24, and 27/28 post exposure (PE) for all NHPs, except for ID CTRL-1, which was sampled at the prescribed schedule until time of euthanasia on day 8 PE. The average absolute lymphocyte (**A**), absolute neutrophil (**B**), absolute monocyte (**C**), and percent monocyte (**D**) are graphed for the control and VRP-vaccinated groups. Gray shaded areas indicate normal range for each analyte.

**Figure 8 viruses-14-01502-f008:**
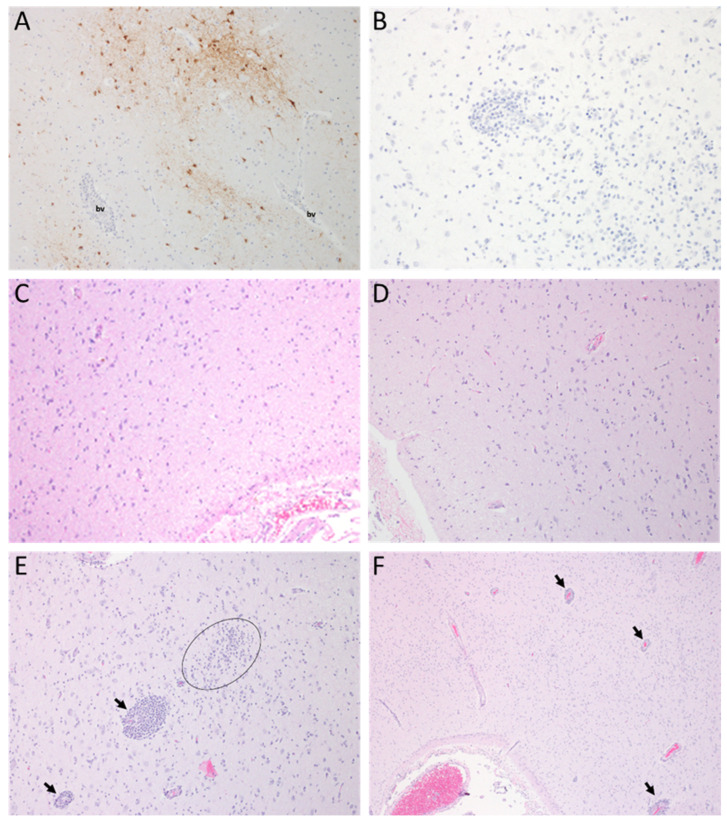
Pathological changes in NHPs following WEEV CBA87 exposure. Tissues were collected at time of euthanasia for immunohistochemistry (**A**,**B**) and hematoxylin and eosin examination (**C**–**F**). All images are of the cerebrum (corpus striatum) region. (**A**) ID CTRL-1 on day 8 post-exposure (PE), positive for WEEV antigen (10×); (**B**) ID CTRL-3 on day 29 PE, no WEEV positive staining (20×); (**C**) Normal healthy tissue (10×); (**D**) ID VRP-6 on day 28 PE (10×); (**E**) IM CTRL-1, perivascular infiltrate (arrows) and gliosis (circle) were observed on day 32 (4×); and (**F**) ID CTRL-1, perivascular infiltrates of lymphocytes and histiocytes (arrows) and multifocal gliosis within the neuropil on day 8 (4×). bv indicates a blood vessel.

**Figure 9 viruses-14-01502-f009:**
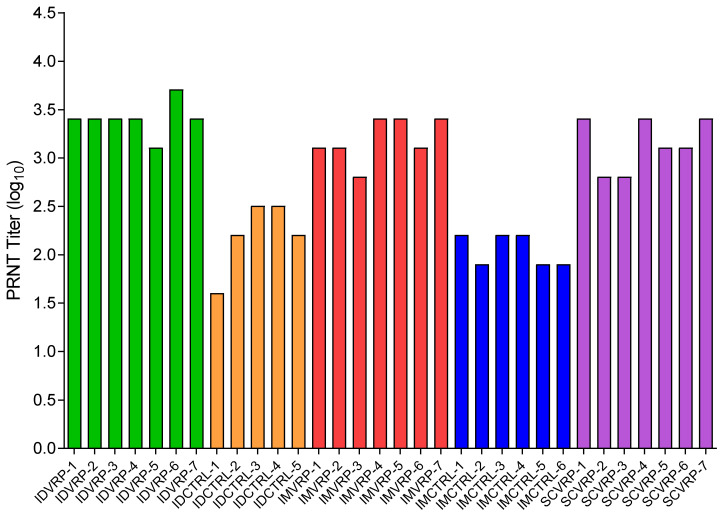
Neutralizing serum antibody titers after WEEV CBA87 aerosol exposure. The PRNT80 titers at the time of death for ID CTRL-1 (day 8 PE) or on day 28 PE for all other NHPs. Blood was collected and the WEEV-specific neutralizing antibody levels were measured by PRNT assay.

**Table 1 viruses-14-01502-t001:** Number of NHPs with Detectable Amplification above Lower Limit of Detection of WEEV CBA87 Viral RNA in the Blood by RT-PCR.

Study Group (*n*)	Day (Relative to Aerosol Exposure to WEEV CBA87)								
	−3	−2	−1	1	2	3	4	5	6
ID VRP (7)	0	0	0	0	0	0	0	0	0
ID CTRL (5)	0	0	0	1 *	3 *	0	0	0	0
IM VRP (7)	0	0	0	0	0	0	0	0	0
IM CTRL (6)	0	0	0	1	2 *	1 **	0	0	0
SC VRP (7)	0	0	0	0	0	0	0	0	0

* One additional NHP had amplification below the Limit of Detection; the sample was repeated with consistent amplification, but at levels under the limits of detection or with high variation; ** Two additional NHPs had amplification below the Limit of Detection; the samples were repeated with consistent amplification, but at levels under the limits of detection or with high variation.

**Table 2 viruses-14-01502-t002:** Number of NHPs with Detectable Amplification above Lower Limit of Detection of WEEV CBA87 Viral RNA in Tissue and Cerebral Spinal Fluid by RT-PCR.

Study Group (*n*)	Tissue Type									
	Mandibular Lymph Node	Tonsil	Lung	Liver	Spleen	Olfactory Bulb	Cerebrum	Cerebellum	Brainstem/ Spinal Cord	Cerebral Spinal Fluid
ID VRP (7)	0	0	0	0	0	0	0	0	0	0
ID CTRL (5)	0	0	0	0	0	3	0	1	1	0 *
IM VRP (7)	0	0	0	0	0	0	0	0	0	0
IM CTRL (6)	0	0	0	0	0	0 *	0	0	0	0
SC VRP (7)	0	0	0	0	0	0	0	0	0	0

* One equivocal sample repeated with variable amplification results.

## Data Availability

The data presented in this study are available on request from the corresponding author. The data are not publicly available as a data sharing agreement is required prior to its release.

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
