# Peer review of "Efficacy of Western, Eastern, and Venezuelan Equine Encephalitis (WEVEE) Virus-Replicon Particle (VRP) Vaccine against WEEV in a Non-Human Primate Animal Model"

_viruses, 2022, doi:10.3390/v14071502_

Round 1

Reviewer 1 Report

In this study, the researchers tested three different routes of vaccination (ID, SC, and IM) for a combined anti-WEEV/VEEV/EEEV VRP-based vaccine. Unfortunately, the Cynomolgus primate model does not provide a "toes up, toes down" readout of vaccine efficacy. The researchers looked at other markers of infection including fever, viremia, viral replication, and pathology to compare vaccinated and control groups. While they report development of neutralizing antibodies and protection from fever in vaccinated animals the dose, route, animal model (or all three) did not permit conclusive findings for protection using other metrics except pathology which showed a dramatic difference between vaccinated animals and controls.

General Comments:

Do you see significant differences if you pool the data from the two control groups?

Why is there no SC-Ctrl group? 

Is there an explanation for the differences in isolation of viral RNA from the two control groups? Do the control groups differ significantly in other ways?

Specific Comments:

Feature the pathology results in the main manuscript, preferably with objective clinical scoring as well as pictures (H&E staining). They do not belong in supplementary data.

Line 587-588: These data demonstrate that the NHPs were productively infected and mounted an immune response.

Seroconversion is evidence of exposure, however, the lack of detectable viral replication or antigen suggests that the animals were not productively infected. It is interesting that encephalitis is present in the control animals without evidence of viral antigen or RNA. This makes it difficult to show that the encephalitis was the result of viral infection, considering that persistence of alphavirus RNA or protein has been observed in other published studies.

Author Response

Do you see significant differences if you pool the data from the two control groups? The Authors are unsure exactly which data set the reviewer is referring to. Statistically significant differences in temperatures between control and vaccinated animals were noted. Comparing the average hematology data, a separation is observed with the absolute lymphocytes and monocytes; however, these never reach statistical significance because of the large NHP to NHP variability within groups. This variability would be more readily visualized with error bars on the graphs. The Authors chose to leave the error bars off those graphs to help with visualizing the data. Due to the lack of error bars, it was important to the Authors to state that statistical significance was not reached. Combining of the two control groups would not reduce the inherit NHP to NHP variability leading to the lack of statistical significance.

Why is there no SC-Ctrl group? The ID and SC inoculations share a control group to help reduce the total number of nonhuman primates needed while maintaining statistical power in groups. By doing so we were being good stewards of animal use and abiding by the 3R’s (refinement, reduction, replacement).

Is there an explanation for the differences in isolation of viral RNA from the two control groups? The RNA from all samples was isolated using the QIAmp Viral RNA kits (Qiagen) as stated in lines 251-252. The Authors are could not identify what portion of the manuscript led the reviewer to think the RNA was isolated differently for the two control groups. If a line number could be provided, the Authors would happily fix the error.

 Do the control groups differ significantly in other ways? The only differences in the control groups beyond placebo administration route is that ID control animals were challenged 56 days after primary placebo administration and IM control animals were challenged 57 days after primary placebo administration. This was necessary due to the number of animals in the study and the time it takes to aerosol expose a NHP. This information is available in the methods section (lines 176-180).

Specific Comments:

Feature the pathology results in the main manuscript, preferably with objective clinical scoring as well as pictures (H&E staining). They do not belong in supplementary data. The Authors appreciate the suggestion and as a result have added Figure 8 pathology images to help demonstrate the findings.

Line 587-588: These data demonstrate that the NHPs were productively infected and mounted an immune response.

Seroconversion is evidence of exposure, however, the lack of detectable viral replication or antigen suggests that the animals were not productively infected. It is interesting that encephalitis is present in the control animals without evidence of viral antigen or RNA. This makes it difficult to show that the encephalitis was the result of viral infection, considering that persistence of alphavirus RNA or protein has been observed in other published studies.

The lack of WEEV viremia observed in the control animals in this study was not unexpected as it has been previously reported that WEEV viremia is not observed in nonhuman primates after either aerosol or subcutaneous challenge (Reed et al 2005, Ko et al 2019, Smith et al 2020).  

“No virus was detected by plaque assay in any blood or throat swab sample obtained after exposure (data not shown).” Reed 2005

“None of the animals infected with WEEV developed a detectable viremia.” Smith 2020

The Authors added Figure 8 Pathology data that we hope will help demonstrate the control NHPs were productively infected thus resulting in the immune response now presented in Figure 9. ID CTRL-1, which was euthanized on Day 8 had positive IHC (Figure 8A) and pathology in the brain by H&E (Figure 8F). Similar pathology was observed in IM CTRL-1 that survived until the Day 32 (end of study; Figure 8E); however, at the end of the study ≥ 28 days after challenge no WEEV antigen was present in the control NHP Cerebrum by IHC (Figure 8B). This data supports the RT-PCR data in Table 2 where viral RNA was only detected in the Cerebrum from ID CTRL-1 NHP that was euthanized on Day 8.

Lack of detection of persistent viral RNA in this study in comparison to other studies may be attributed to 1) this study is examining WEEV in comparison to EEEV or VEEV and the viruses manifest disease in different ways, or 2) our methods of RT-PCR and IHC are less sensitive for the detection of viral antigen than others.   

Reviewer 2 Report

The objective of the study “Efficacy of Western, Eastern, and Venezuelan Equine Encephalitis (WEVEE) Virus-Replicon Particle (VRP) Vaccine against WEEV in a Non-human Primate Animal Model” authored by C.W.Burke et al., was evaluation of the effects of the route of administration on the immunogenicity and efficacy of a combined western, eastern, and Venezuelan equine encephalitis (WEVEE) virus-like replicon particle (VRP) vaccine in NHP.

Since there are no FDA-approved vaccines against VEEV, WEEV or EEEV, development of new, safe and efficacious vaccine candidates against these viruses remains of significant importance for the public health. Virus-like replicon particles (VRPs) represent a perspective approach to generation of such vaccines.

This study represents a continuation of the previous work conducted by the same group on mice and cynomolgus macaques, and provides new data addressing the needs unmet in the previous studies.  Materials and methods used in this study are relevant and adequate to the objectives.

The vaccine used in the current study, consisted of equal amounts of WEEV, EEEV, and VEEV VRPs. Concurrent immunization with all 3 different VRPs allows to avoid any potential immune interference, and induce immunity against all 3 viruses. Significant immune responses were detected in immunized animals to all 3 species of viruses. Three different routes of immunization (intradermal (IM), subcutaneous (SC) and intradermal (ID)) were compared using multiple methods and parameters.

Specific comments and questions.

All three types of VRPs, used in the study, were manufactured and titered by AlphaVax in 2010. What is known about stability of infectious VRP particles, and were they re-titered for use in this study?

Please consider making small edits/corrections throughout the manuscript, suggested below.

Line 205: insert “agarose” before 0.6%

Line 215: replace 6 ug/ml with 6 µg/ml

Line 216: insert space after 300 in “300µg”

Line 234: please specify the log base in “serially log diluted”

Line 249: insert “of” before “blood”

Line 251: consider replacing “specifications” with “instructions”

Line 539: please consider using “27/28” in place of “28” to stay consistent with Line 553.

Line 607: please consider replacing “sites” with “routes”

Table 1. Please double check/explain better about numbers marked up with asterisks. It seems the legend contradicts the Table content.

Figure 8. Please double check if IMCTRL-1 is represented by a red bar, or should it be a blue one?

Author Response

All three types of VRPs, used in the study, were manufactured and titered by AlphaVax in 2010. What is known about stability of infectious VRP particles, and were they re-titered for use in this study? Unfortunately the VRP particles were not re-titrated prior to use on this study. They were manufactured in 2010 and the study was completed in 2014. We speculate that any decay in potency of a particular VRP (VEEV, EEEV, or WEEV) would likely occur at equivalent rates as the preparations were stored in the same location but cannot say with certainty that is the case.  

Please consider making small edits/corrections throughout the manuscript, suggested below.

Line 205: insert “agarose” before 0.6% Done

Line 215: replace 6 ug/ml with 6 µg/ml Done

Line 216: insert space after 300 in “300µg” Done

Line 234: please specify the log base in “serially log diluted” Added information

Line 249: insert “of” before “blood” Done

Line 251: consider replacing “specifications” with “instructions” Done

Line 539: please consider using “27/28” in place of “28” to stay consistent with Line 553. Done

Line 607: please consider replacing “sites” with “routes” Done

Table 1. Please double check/explain better about numbers marked up with asterisks. It seems the legend contradicts the Table content. Updated the footnotes to clarify

Figure 8. Please double check if IMCTRL-1 is represented by a red bar, or should it be a blue one? Fixed

Round 2

Reviewer 1 Report

I recommend acceptance of this manuscript. WEEV continues to be understudied and this research would contribute to the overall body of literature.